# Topical Antibiotic Prophylaxis and Intravitreal Injections: Impact on the Incidence of Acute Endophthalmitis—A Nationwide Study in France from 2009 to 2018

**DOI:** 10.3390/pharmaceutics14102133

**Published:** 2022-10-07

**Authors:** Florian Baudin, Eric Benzenine, Anne-Sophie Mariet, Inès Ben Ghezala, Alain M. Bron, Vincent Daien, Pierre-Henry Gabrielle, Catherine Quantin, Catherine Creuzot-Garcher

**Affiliations:** 1Department of Ophthalmology, University Hospital, 21000 Dijon, France; 2EA7460, PEC2, Cerebral and Cardiovascular Epidemiology, and Physiopathology, 21000 Dijon, France; 3Biostatistics and Bioinformatics (DIM), University Hospital, 21079 Dijon, France; 4Biostatistics and Bioinformatics (DIM), University Hospital, Bourgogne Franche-Comté University, BP 77908, 21079 Dijon, France; 5INSERM, CIC 1432, 21000 Dijon, France; 6Clinical Epidemiology, Clinical Trials Unit, Clinical Investigation Center, Dijon University Hospital, 21000 Dijon, France; 7Biostatistics, Biomathematics, Pharmacoepidemiology and Infectious Diseases (B2PHI), INSERM, UVSQ, Institut Pasteur, Université Paris-Saclay, 94807 Paris, France; 8Eye and Nutrition Research Group, Bourgogne Franche-Comté University, 21000 Dijon, France; 9Department of Ophthalmology, University Hospital, 34295 Montpellier, France; 10Inserm U1061, University of Montpellier, 34000 Montpellier, France

**Keywords:** IVT, intravitreal injection, endophthalmitis, antibiotics

## Abstract

***Background:*** The dramatic increase in intravitreal injections (IVTs) has been accompanied by a greater need for safer procedures. The ongoing debate about topical antibiotic prophylaxis after IVTs emphasizes the importance of large-scale studies. We aimed to study the role of topical antibiotic prophylaxis in reducing the risk of acute endophthalmitis after IVTs. ***Methods:*** Population-based cohort study, in France, from 2009 to 2018, including all French patients receiving IVTs of corticosteroids or anti-VEGF agents. ***Results:*** A total of 5,291,420 IVTs were performed on 605,434 patients. The rate of topical antibiotic prophylaxis after IVTs progressively decreased during the study period, with a sharp drop in 2014 (from 84.6% in 2009 to 27.4% in 2018). Acute endophthalmitis occurred in 1274 cases (incidence rate = 0.0241%). Although antibiotic prophylaxis did not alter the risk of endophthalmitis (*p* = 0.06), univariate analysis showed an increased risk after fluoroquinolone and aminoglycoside prophylaxis. This increased risk was not found in multivariate analysis. However, we observed an increased risk related to the use of fixed combinations of fluoroquinolones and aminoglycosides with corticosteroids (IRR = 1.89; 95% CI = 1.57–2.27%, antibiotics combined with corticosteroids). ***Conclusion:*** These results are consistent with the literature. Endophthalmitis rates after IVTs did not decrease with topical antibiotic prophylaxis. The use of a combination of antibiotics and corticosteroids doubles the risk of endophthalmitis and should be avoided. Avoiding antibiotic prophylaxis would reduce the costs and the potential risks of antibiotic resistance.

## 1. Introduction

Intravitreal injections (IVTs) have dramatically improved the visual prognosis of patients suffering from various ocular conditions, such as age-related macular degeneration, macular edema linked to diabetes, and retinal vein occlusion. Consequently, a rapidly growing number of patients have been treated with anti-VEGF (vascular endothelial growth factor) agents or corticosteroids. In France, more than 500,000 injections are administered to 150,000 patients annually [1]. One of the most dreaded complications after IVTs is acute endophthalmitis, which occurs at a low rate of 0.0245% [1]. Reducing the rate of endophthalmitis is of major concern. However, there is little evidence that the administration of topical antibiotics reduces the rate of endophthalmitis after IVTs. To date, the use of povidone-iodine on the ocular surface has been the only procedure proven effective in reducing the risk of endophthalmitis [2,3]. Given the lack of evidence of increased protection with antibiotic prophylaxis compared to no antibiotics [3,4,5], the French guidelines were modified in 2014, and topical antibiotic prophylaxis was no longer recommended [6]. We sought to determine whether antibiotic prophylaxis plays a role in the incidence rate of acute endophthalmitis following IVTs on a national scale. To the best of our knowledge, this is the only countrywide study on this question. Due to the low rates of endophthalmitis and the different topical antibiotics used, a large cohort study over several years was required. In the present study, we aimed to examine the role of antibiotic prophylaxis in reducing the risk of acute endophthalmitis after IVTs in France from 2009 to 2018.

## 2. Materials and Methods

### 2.1. Data Source

This study is part of the French Epidemiology and Safety (EPISAFE) collaborative program [7]. This project emerged from recognizing the limitations of current epidemiological or interventional studies in determining the effects of different ophthalmic procedures, particularly with regard to rare events. The French National Health Data System (SNDS) was created in 2016 to develop health data and represents a significant advance in analyzing and improving population health [8]. Managed by the French National Health Insurance Fund (*Caisse Nationale de l’Assurance Maladie*, CNAM), the SNDS includes data from the *Assurance Maladie* (database of the national inter-organizational information system of the Assurance Maladie, SNIIRAM), hospital data (database of the Program for the Medicalization of Information Systems—PMSI), and databases on deaths and disability, collecting data for the whole country, i.e., 66 million inhabitants. The high quality of this database has previously been evaluated and used in several epidemiological studies [9,10,11,12]. The present study was approved by the French Institute of Health Data (registration number 115306, 24 January 2019) and the French Data Protection Authority (registration number D.R. 2019-100, 12 April 2019). This study adhered to the tenets of the Declaration of Helsinki. The study was registered on the Clinical Trials site under the reference NCT03635268 (https://clinicaltrials.gov/ct2/show/NCT03635268 (accessed on 5 September 2022)).

### 2.2. Data Extraction

The dataset for this study included all adult patients in the database who received at least one IVT from 1 January 2007 to 31 December 2018. Data were not included if a lookback period or follow-up of 42 days was unavailable or if the patient died within the 42-day follow-up period. Procedures performed before 2009 were used to verify the uniqueness of cause, medical history, and medicine consumption, since the exact dates of hospitalization were unavailable for 2007 and 2008 (see Appendix A). As a result, only index dates between 12 February 2009 and 19 November 2018 were considered. IVTs were tracked with the billing code for IVT (BGLB001, Common Classification of Medical Procedures) used as the index date. Endophthalmitis cases were tracked with the billing codes H440 or H441 (*Purulent endophthalmitis* and *Other endophthalmitis*, respectively; 10th edition of the *International Classification of Diseases*) within 42 days of the injection index date [13]. Cases of endophthalmitis following other surgeries, other exogenous and traumatic causes, or endogenous causes secondary to infectious pathologies were excluded. Data on the patient’s sex, age at the beginning of treatment, information on injections, endophthalmitis diagnosis, and diabetic and insulin-dependent status were collected based on hospitalization discharge codes, long-term disease, and chronic treatments [14]. Information on the injections included the therapeutic class, the drug injected, whether the syringe was prefilled or not, the use of antibiotic prophylaxis and the therapeutic class, and whether it was a fixed combination of antibiotic and topical corticosteroid. The drugs injected during the study period were ranibizumab (0.5 mg/0.05 mL; Lucentis^®^; Novartis Pharma SAS, Basel, Switzerland), bevacizumab (1.25 mg/0.05 mL; Avastin^®^; Roche, Basel, Switzerland), aflibercept (2 mg/0.05 mL; Eylea^®^; Bayer HealthCare, Berlin, Germany), pegaptanib (0.3 mg; Macugen^®^, Pfizer, Inc., Eyetech Pharmaceuticals Inc., New York, NY, USA), triamcinolone acetonide (4 mg/0.1 mL; Kenacort^®^; Bristol-Myers Squibb, New York, NY, USA), and dexamethasone implants (0.7 mg; Ozurdex^®^; Allergan SAS, Irvine, CA, USA). To study the impact of the date of injection on the occurrence of endophthalmitis in multivariate analysis, we considered the periods before and after 2014, in relation to the univariate results of lower IRRs after 2014 and to the authorization of prefilled syringes of ranibizumab as well as the recommendations of the French Society of Ophthalmology in 2014 regarding the lack of benefit of antibiotic prophylaxis after IVT [6].

### 2.3. Statistical Analysis

According to the results of a Kolmogorov–Smirnov normality test [15], most of the continuous variables did not follow a normal distribution. Therefore, medians and interquartile ranges (IQRs) were provided for continuous variables, and nonparametric tests were used for comparison. Numbers and percentages were provided for categorical variables, and the chi-squared test was used to compare percentages. We estimated incidence rates as the number of events per 100 IVT procedures. Incidence rate ratios (IRRs) were estimated with Poisson regressions. Using a univariate Poisson regression, we first analyzed the relationships between the variables and the occurrence of endophthalmitis. Multivariate Poisson regressions were then performed, adjusting for the following potential confounders: sex, age, diabetes, drugs, drug preparation, and topical antibiotic prophylaxis. Analyses were based on repeated-measures Poisson regression models accounting for dependencies between repeated observations on the same subject and collinearity between concurrent antibiotic prophylaxes. These models estimated the association between the variables studied and the outcomes using IRRs and the corresponding 95% confidence intervals (CIs). The selection of the multivariate model was based on statistically associated covariates in univariate regression and comparing several candidate models based on the quasi-likelihood under the independence model criterion (QIC) proposed by Pan, used to compare generalized estimating equation (GEE) models [16]. Statistical significance was set at *p* < 0.05 (two-tailed tests). Data processing and statistical analyses were performed using the SAS statistical analysis software package (SAS Enterprise Guide^®^ version 7.1; and SAS version 9.4; SAS Institute, Inc., Cary, NC, USA).

## 3. Results

From 2009 to 2018, a total of 5,291,420 IVTs were performed and analyzed in the present study, after excluding IVTs with insufficient lookback or lacking 42 days of follow-up, as well as IVTs concomitant with ocular surgery (Table 1). 

Continuous variables are displayed as medians and interquartile ranges. Categorical variables are displayed as numbers and percentages.

IVTs were performed on 605,434 patients aged 78.0 years at treatment initiation (IQR, 68.0–84.0), and most of them were women (58.8%). Most IVTs were anti-VEGF injections, accounting for 91.0% of all procedures; 4.1% were corticosteroids, and 4.9% were not identified in the database. The most frequently injected agent was ranibizumab (63.1% of all injections), followed by aflibercept (27.5%) (Figure 1 and Table 2).

The years 2009 and 2018 were truncated to allow for the required 42 days of lookback and follow-up.

Topical antibiotic prophylaxis was given in 57.0% of all injections, with the most prescribed antibiotic class being macrolides (57.0%), followed by aminoglycosides (21.8%) and fluoroquinolones (20.3%). Combination medications with antibiotics and corticosteroids were administered in 5.4% of the IVTs. Corticosteroid IVTs were most likely performed with topical antibiotic prophylaxis (60.8% vs. 57.5% for anti-VEGF, *p* < 0.001). The prevalence of IVTs performed with antibiotic prophylaxis decreased during the study period (Figure 2). While 84.6% of IVTs were administered together with antibiotic prophylaxis in 2009, this rate has progressively decreased, especially since the publication of recommendations in 2014 discouraging antibiotic prophylaxis (2014: 75.1%, 2015: 56.9%, 2018: 27.4%).

We recorded 1274 endophthalmitis cases out of 5,291,420 IVTs (1/4153 injections, 0.0241%) during the study period. The incidence of endophthalmitis following anti-VEGF and corticosteroid injections was 0.0197% and 0.0699%, respectively. Figure 3 displays the proportion of IVTs with antibiotic prophylaxis and the endophthalmitis rates per year.

Endophthalmitis occurred mostly after fluoroquinolone, aminoglycoside, and macrolide antibiotic prophylaxes, owing to their more frequent prescription (*n* = 189, 222, and 439, respectively). No cases or few cases of endophthalmitis were observed under less frequently prescribed antibiotic prophylaxes such as phenicols, polypeptides, fusidic acid, or rifamycin (*n* = 0, 1, 2, and 8, respectively). In contrast, a higher number of endophthalmitis cases was observed for more frequently prescribed antibiotic prophylaxes such as fluoroquinolones, aminoglycosides, and macrolides (*n* = 189, 222, and 439, respectively). No statistically significant differences were found for sex and diabetes when considering IVTs with or without endophthalmitis. Patients with endophthalmitis were significantly younger than those without it (78 years (70–84) vs. 79 (71–85), *p* < 0.001). Endophthalmitis occurred earlier with antibiotic prophylaxis than without: 6.6 days (±7.8) vs. 7.8 (±8.6), respectively (*p* = 0.01). In univariate analysis, acute post-IVT endophthalmitis was more likely to occur in younger patients, with corticosteroid IVTs, with non-prefilled anti-VEGF syringes (vs. prefilled ranibizumab), and at the beginning of the study period, with a decreasing IRR over the years (Figure 4).

There was no association when the role of antibiotic prophylaxis in the occurrence of endophthalmitis was examined in univariate analysis, regardless of the type of antibiotic, while antibiotic prophylaxis by aminoglycosides or fluoroquinolones yielded a greater risk of endophthalmitis than no prophylaxis (IRR, 1.36; 95% CI, 1.18–1.57%; *p* < 0.001; IRR, 1.22; 95% CI, 1.05–1.42%; *p* = 0.02, respectively). Other antibiotics did not decrease or increase the risk of post-IVT endophthalmitis. A major finding was that IVT with a topical combination of antibiotics and corticosteroids was associated with a higher risk of endophthalmitis versus no prophylaxis (incidence of 0.044%, IRR, 1.89; 95% CI, 1.57–2.27%; *p* < 0.001), independent of the therapeutic class injected. Other variables positively associated with a higher occurrence of endophthalmitis included corticosteroid IVTs, with a significant risk after dexamethasone and triamcinolone injection (IRR, 2.83; 95% CI, 2.30–3.50%, and 10.83; 7.69–15.26%, respectively). Variables associated with a lower likelihood of endophthalmitis were older age, prefilled vs. non-prefilled ranibizumab, and period after 2014 (except 2018) (*p* < 0.001).

In multivariate analysis, after the variable selection process, post-IVT acute endophthalmitis was more likely to occur with corticosteroid IVTs of dexamethasone and triamcinolone (IRR, 2.86; 95% CI, 2.31–3.53%, and 10.23; 7.25–14.43%, respectively) or with a combination of topical aminoglycoside and corticosteroids (IRR, 1.50; 95% CI, 1.21–1.86%) (Figure 4). The increased risk of endophthalmitis found in univariate analysis after prophylaxis with aminoglycosides or fluoroquinolones was not found in multivariate analysis, but only for combined formulations of antibiotics and corticosteroids. Furthermore, antibiotic prophylaxis alone did not modify the risk of endophthalmitis. Endophthalmitis was less likely to occur in males and with prefilled ranibizumab than non-prefilled ranibizumab.

## 4. Discussion

In this study examining all IVTs performed in France over a period of almost 10 years, we observed a low post-injection endophthalmitis rate of 0.0241% (1/4153 injections). This rate is consistent with other reports, ranging from 0.02% to 0.08% [1,17,18,19,20]. As previously reported, an association was found between endophthalmitis incidence and the type of drug injected as well as the preparation used, with corticosteroids and non-prefilled syringes linked to a higher risk of endophthalmitis [1,21,22]. Antibiotic prophylaxis alone did not modify the risk of endophthalmitis; however, an increased risk related to the use of fixed combinations of fluoroquinolones and aminoglycosides with corticosteroids was observed.

There is a long-running debate on the effectiveness of antibiotic prophylaxis after IVTs. None of the previously published data were sufficiently powered to reach a robust conclusion. Moreover, published studies suggested that patients with endophthalmitis were at higher risk of bacterial resistance following antibiotic prophylaxis [23,24], due to the modification and selection of the conjunctival flora by repeated antibiotic treatments [25]. In the literature, topical antibiotics have been reported to increase the risk of resistance and the minimum inhibitory concentration of strains of the conjunctival flora [24]. Topical antibiotics cannot reach effective aqueous humor or intravitreal concentrations for a bactericidal, therapeutic effect [26,27]. In a randomized study of eyes treated with topical antibiotics after repeated injections, the rate of fluoroquinolone resistance reached 67–85% after 1 year [28]. Our previous study of 316,576 IVTs in France reported an overall endophthalmitis rate of 0.021%, in which antibiotic or antiseptic prophylaxis was associated with increased rates of endophthalmitis in both univariate (*p* = 0.02) and multivariate (*p* = 0.001) analyses [20]. In 2016, Benoist d’Azy et al. conducted a meta-analysis that did not find a difference in the risk of endophthalmitis after antibiotic prophylaxis [29], while Menchini et al. found that it could increase the risk of endophthalmitis [30]. As a result, the proportion of IVTs performed with antibiotic prophylaxis decreased with time, and national guidelines were modified. In France, as of May 2014, a topical antibiotic is no longer recommended following anti-VEGF IVTs [6]. In our study, no significant association was found between endophthalmitis after IVTs and the use of topical antibiotic prophylaxis in our cohort, except for corticosteroid-associated antibiotics, which were shown to be associated with a higher risk of endophthalmitis after anti-VEGF IVTs in univariate analysis. Indeed, the combination of antibiotics and corticosteroids almost doubles the risk of endophthalmitis (IRR = 1.89). Their immunosuppressive properties could explain this higher risk [5,31], increasing the probability of bacterial or fungal endophthalmitis [32].

The French guidelines were amended in 2014, resulting in a significant reduction in post-IVT antibiotic prophylaxis, as shown in our study (see Figure 3), without any subsequent increase in the risk of endophthalmitis. Our results are consistent with the observations made by Torres-Costa et al. [33]. Retinal specialists have changed their practice, with a recent survey revealing that only 10.9% use topical antibiotics before injection and only 16.6% do so after injection—a substantial change when compared to a similar survey in 2011 [34,35].

### 4.1. Limitations

We recognize some limitations in this study, which are partly inherent to medical-administrative database studies.


First, concerning the event under study, acute post-IVT endophthalmitis was defined by the record of a hospital stay for endophthalmitis. In France, the practice is to admit patients with endophthalmitis to hospital. In addition, the consumption of care (i.e., procedures, hospitalizations, and medication) was verified in those cases. Finally, although we could not obtain laboratory confirmation of the diagnosis, it is well documented that this piece of information is missing in a substantial number of cases [36]. Moreover, the incidence found in our national cohort was very close to that found in the literature [20,37,38].Second, we could not determine precisely whether the topical antibiotic prophylaxis was given before or after the IVTs, or its duration. However, this did not appear to modify the risk of endophthalmitis [29,39].Third, the potential role of patient adherence to their drops once prescriptions were delivered may have to be taken into consideration.Fourth, we had no information on the asepsis protocol. However, as reported by Dossarps et al., there is a unique asepsis protocol that is used exclusively in France [20,40].Fifth, we could not adjust our data to systemic or local infectious risk factors such as immunosuppression, except for diabetic status or conjunctiva at risk.Sixth, we limited our main outcome measures to infectious events occurring in the 42 days following the procedure, as defined by the Endophthalmitis Vitrectomy Study Group (EVS) [13]. However, our results were consistent with previous findings that 90% of endophthalmitis cases occurred within the first 2 weeks.Seventh, our findings cannot be fully extended to other countries; French guidelines for IVTs are somewhat different from American guidelines [40,41]. In France, performing IVTs in a dedicated room wearing sterile gloves is recommended. There does not appear to be any difference in the risk of endophthalmitis depending on the setting of the injection [42] and the use of sterile gloves or not [43,44]. In contrast, similar recommendations in these two countries include topical povidone-iodine use, surgical mask wear, and the absence of topical antibiotics [30,40,44]. Povidone-iodine is the only antiseptic technique that has been proven to decrease the risk of endophthalmitis after IVTs [45]. In France, the standard practice is to use povidone-iodine for all patients. Finally, the conclusions drawn from big data must be interpreted cautiously due to their limitations, as has already been pointed out in the ophthalmic literature [46]. While numerous potential confounders were adjusted in the analyses, other confounders not included or studied could be associated with the risk of endophthalmitis.


### 4.2. Strengths

The strength of this study is the collection of all IVTs registered in a single administrative database over 10 years in our country.


The subgroup size was large enough to detect a statistically significant difference between exposure groups;In a quasi-exhaustive population;Without exclusion criteria as found in RCTs or selected population studies (e.g., Medicare studies).


### 4.3. Conclusions

In conclusion, this study found that antibiotic prophylaxis for IVTs does not lower the risk of endophthalmitis and could be detrimental through the selection of resistant germs in the conjunctival flora, thereby leading to more aggressive endophthalmitis cases, as reported in the literature. In addition, using a combination of antibiotics with corticosteroids doubles the risk of endophthalmitis and should be avoided. Patients receiving anti-VEGF or corticosteroid IVTs should therefore not be given antibiotic prophylaxis.

## Figures and Tables

**Figure 1 pharmaceutics-14-02133-f001:**
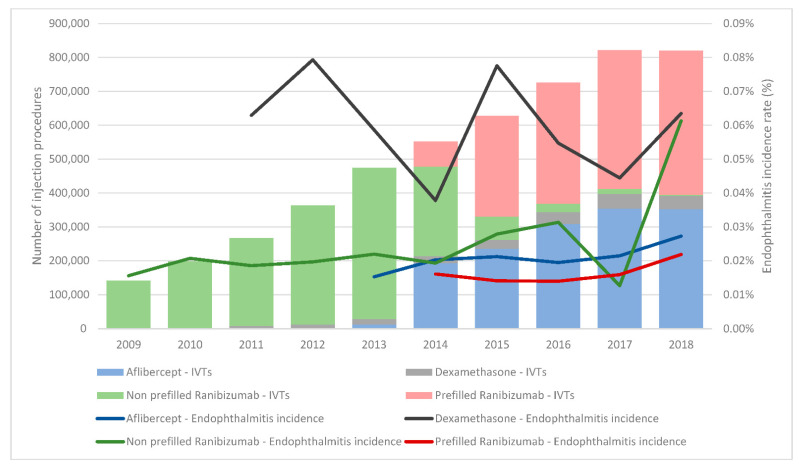
Proportion of IVT treatments and their relative endophthalmitis incidence rates. Due to the low proportion of IVTs with triamcinolone, pegaptanib, and bevacizumab, they are not displayed in this figure.

**Figure 2 pharmaceutics-14-02133-f002:**
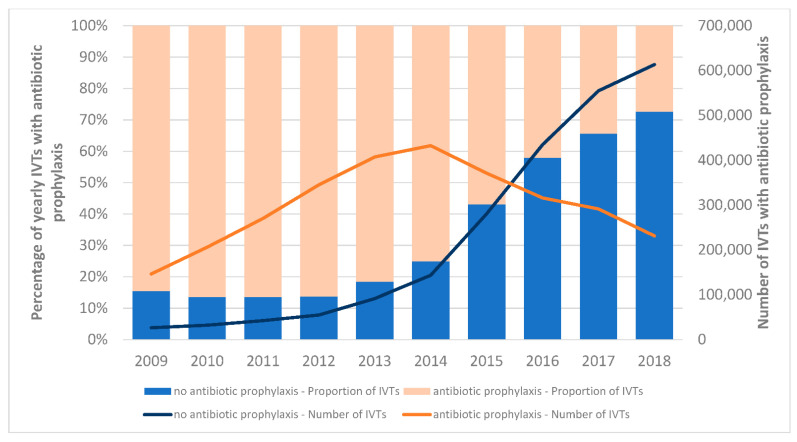
Changes in antibiotic prophylaxis use with IVTs from 2009 to 2018. The years 2009 and 2018 are truncated to allow for the required 42 days of lookback and follow-up.

**Figure 3 pharmaceutics-14-02133-f003:**
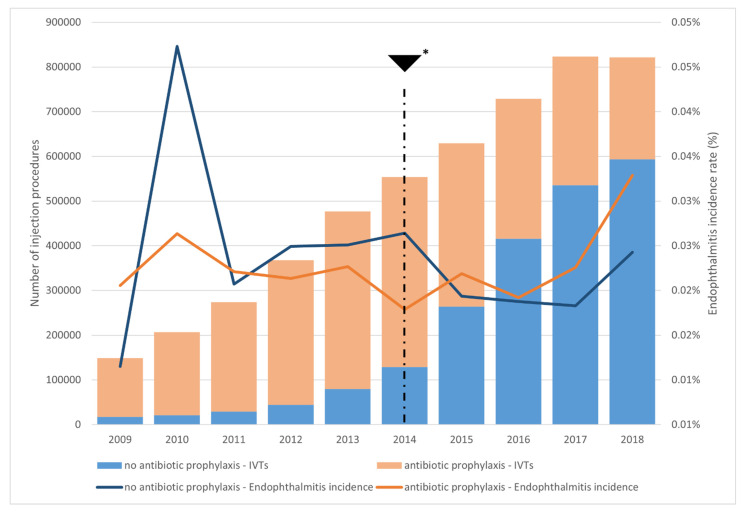
Proportion of IVTs performed with antibiotic prophylaxis per year, and corresponding endophthalmitis rates per year; * 2014, 2nd quarter, publication of the French guidelines on the absence of any benefit from antibiotic prophylaxis and prefilled ranibizumab availability. The years 2009 and 2018 are truncated to allow for the required 42 days of lookback and follow-up.

**Figure 4 pharmaceutics-14-02133-f004:**
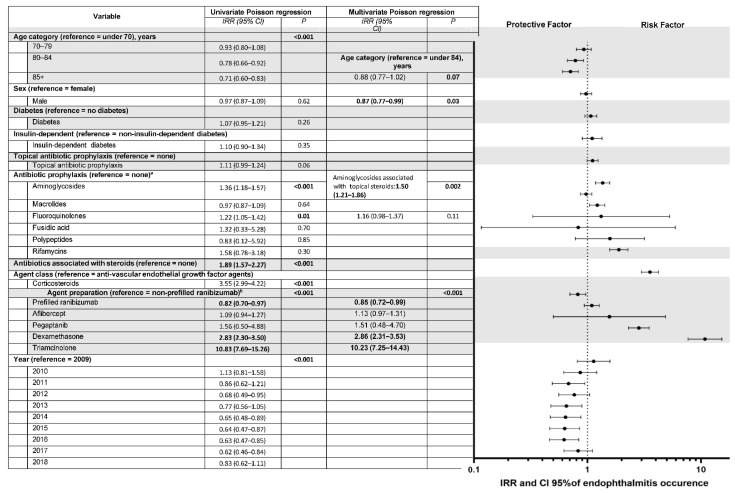
Univariate and multivariate analysis of factors associated with acute endophthalmitis after intravitreal injections of corticosteroids or anti-VEGF (vascular endothelial growth factor) agents from 2009 to 2018. IRR = incidence rate ratio; CI = confidence interval. Missing data for agent class and specific agent, *n* = 259,802. Multivariate Poisson regression with variable selection based on significant association with the event and on the quasi-likelihood under the independence model criterion (QIC) proposed by Pan, used to compare GEE models [16]. ^a^ Since we recorded only one prescription of phenicol antibiotics, univariate analysis is not shown for this class. ^b^ Since no endophthalmitis occurred after intravitreal bevacizumab injections, these 13,562 injections were not considered for the by-agent analysis.

**Table 1 pharmaceutics-14-02133-t001:** Baseline demographics of patients with intravitreal injections of corticosteroids or anti-VEGF (vascular endothelial growth factor) agents from 2009 to 2018.

Characteristics	Patient Data (*n* = 605,434)
Age, years	79 (69–85)
Sex, female	356,196 (58.8%)
Number of injections	5 (3–11)
Follow-up, days	304 (61–1035)
Patients with diabetes, *n* (%)	187,918 (31.0%)
Insulin-dependent patients with diabetes, *n* (%)	93,556 (49.8%)

**Table 2 pharmaceutics-14-02133-t002:** Proportion of IVT treatments.

Therapeutic Class	Agent	Frequency	Overall Proportion
**Anti-VEGF**	**Bevacizumab**	13,567	0.26%
	**Aflibercept**	1,455,218	27.50%
	**Ranibizumab**	3,337,135	63.07%
	**Pegaptanib**	9435	0.18%
**Corticosteroid**	**Triamcinolone**	16,365	0.31%
	**Dexamethasone**	199,751	3.77%
**Unknown**		259,949	4.91%
	**TOTAL**	5,291,420	

## Data Availability

Specific remote access to anonymized Medicare data was granted by the French Institute of Health Data and by the French Data Protection Authority; these data are not accessible.

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
