# Peer review of "Topical Antibiotic Prophylaxis and Intravitreal Injections: Impact on the Incidence of Acute Endophthalmitis—A Nationwide Study in France from 2009 to 2018"

_pharmaceutics, 2022, doi:10.3390/pharmaceutics14102133_

Round 1
Reviewer 1 Report
Dear Author,
Very important and interesting manuscript, but we need improve article.
1) Article need more information about inclusion and exlusion criteria
2) If possible improve design of trial (probably good idea make graphic scheme)
3) Provide more clinical advice in conclosion
Author Response
We thank reviewer 1 for his insight. You should find attached our response to both reviewers comments.
Yours sincerely,

Reviewer 2 Report
Florian Baudin and co-workers examined the role of antibiotic prophylaxis in reducing the risk of acute endophthalmitis after IVTs in France from 2009 to 2018. Further, they studied whether antibiotic prophylaxis had a role in the incidence rate of acute endophthalmitis following IVTs on a national scale. The authors clearly stated the strengths and limitations of their study. This study found that antibiotic prophylaxis for IVTs does not lower the risk of endophthalmitis.
Overall, the presentation and datasets are clear and well presented. This work will have a significant impact on real-world clinical settings and well deserved to be published in the Pharmaceutics journal. This paper adds and advances the existence of knowledge in the field.
However, I have the following comments, which must be addressed before acceptance in the journal.
Introduction:
“Considering this lack of evidence [3-5], French guidelines were modified in 2014, and topical antibiotic prophylaxis was no longer recommended [6].”
1. Please elaborate on this lack of evidence in detail. What made French authorities take such a decision? Include this manuscript.
“One of the most dreaded complications after IVTs is acute endophthalmitis, which occurs at a low rate of 0.0245% [1].”
2. Acute endophthalmitis occurred in 1,274 cases (incidence rate = 0.0241%). If your results are close to the existing reference emphasize this fact in the discussion part.
3. The Authors did not mention if any other country study is available and the data. Ste this clearly.
Data Source
1. Add a link to your clinical trial in the manuscript https://clinicaltrials.gov/ct2/show/NCT03635268
Data Extraction
1. “The drugs injected during the study period were ranibizumab (0.5mg/0.05 mL; Lucentis®; Novartis Pharma SAS, Basel, Switzerland), bevacizumab 1.25mg/0.05 mL; Avastin®; Roche, Basel, Switzerland), aflibercept (2 mg/0.05 mL; Eylea®; Bayer HealthCare, Berlin, Germany), pegaptanib (0.3 mg; Macugen®, Pfizer, Inc., Eyetech Pharmaceuticals Inc., USA), triamcinolone acetonide (4 mg/0.1 mL; Kenacort®; Bristol-Myers Squibb, New York, NY, USA), and dexamethasone implant (0.7 mg; Ozurdex®; Allergan SAS, Irvine, CA, USA).”
Provide a table for the number of injections for each molecule. Also mention if the patient number is available in parentheses.
Statistical Analysis
1. Provide a reference for Kolmogorov–Smirnov normality test.
2. P<.05 avoid this and add zero in front of point 0.05 everywhere in the manuscript.
Results
1. So according to your data, there was no patient below 69 years of age?
2. Topical antibiotic prophylaxis was given in 57.0% of all injections, the most prescribed antibiotic class being macrolides (57.0%), followed by aminoglycosides (21.8%) and fluoroquinolones (20.3%).
Mention these antibiotics' names for better clarity.
3. Reference error line 172
Discussion:
1. 270 line reference error.
2. 276-296 lines this is an excellent section. Convert it into subheadings and make it bullet points or sub-numbered points. You are masking its impact by writing it as a paragraph.
3. 296-304 discuss the data available in different countries for the same observations.
4. Make strengths as bullet points.
Datasets:
I kindly request authors to deposit the data in supplementary files because it will have a wide range of utility beyond this study, especially IVT injections, implants, and novel drug delivery systems. The least I expect is to deposit molecule-specific IVT injection number data for each marketed formulation.
Conflicts of Interest:
Some of the authors are consultants for the companies so they must clearly state “The company had no role in the design of the study; in the collection, analyses, or interpretation of data; in the writing of the manuscript, or in the decision to publish the results.”
They should also disclose commercial interests such as holding stocks etc.
Author Response
We thank reviewer 2 for his insight. You should find attached our response to both reviewers comments.
Yours sincerely,

Round 2
Reviewer 2 Report
I accept the revised version in its current form.